# MGTST: Multi-scale and Cross-channel Gated Transformer for Multivariate long-term time-series forecasting

## Abstract

Transformer-based models have emerged as popular choices for the multivariate long-term time-series forecasting problem due to their ability to capture long-term dependencies. However, current transformer-based models either overlook crucial mutual dependencies among channels or fail to capture various temporal patterns across different scales. To fill the gap, we propose a novel model called MGTST (Multi-scale and cross-channel Gated Time-Series Transformer). In this model, we introduce three innovative designs, including Parallel Multi-Scale Architecture (PMSA), Temporal Embedding with Representation Tokens (TERT), and Cross-Channel Attention and Gated Mechanism (CCAGM). In addition, we introduce Channel Grouping (CG) to mitigate channel interaction redundancy for datasets with a large number of channels. The experimental results demonstrate that our model outperforms both channel-dependent (CD) and channel-independent (CI) baseline models on seven widely used benchmark datasets, with performance improvement ranging from 1.5 percent to 41.9 percent when compared to the current state-of-the-art models in terms of forecasting accuracy.

## 1 Introduction

Time-series forecasting is a vital task in time-series analysis, as it involves predicting future observations based on historical time-series data. Accurate predictions are crucial for real-life applications such as weather forecasting (Murphy, 1993), traffic forecasting (Lana et al., 2018), and stock price forecasting (Mondal et al., 2014). Multivariate long-term time-series forecasting (MLTSF) is an even more complex and meaningful task, requiring models to predict the relatively long-term future of the time series with multiple variables. For instance, accurately forecasting weather conditions across multiple locations for the upcoming week enables collaborative precautions against extreme weather events. Deep learning models (Tokgöz & Ünal, 2018; Xue et al., 2019; Li et al., 2019) have emerged as highly effective tools in the realm of MLTSF. Among them, transformer-based models have emerged as highly promising approaches for time-series forecasting, primarily due to their exceptional ability to capture long-term temporal dependencies (Vaswani et al., 2017). Furthermore, several transformer-based models are proposed to reduce computational complexity (Zhou et al., 2021) or enhance forecasting accuracy (Wu et al., 2021).

Despite these advancements, the current performance of transformer models in MLTSF still falls short of expectations. One prominent limitation pertains to the **inadequate modeling of cross-channel dependencies**, which refers to the interrelationships among different variables that have the potential to enhance prediction accuracy. Existing transformer-based models often adopt the channel embedding strategy, which embeds multiple channels at the same time point into a vector representation. This approach yields inferior performance compared to even rudimentary linear models (Zeng et al., 2023). One potential explanation is that this approach is sensitive to the distribution shift between the training set and the test set (Han et al., 2023). Furthermore, certain recent works have neglected cross-channel dependencies altogether (Nie et al., 2022; Wu et al., 2022). Nevertheless, we contend that the performance of current transformer-based MLTSF models can be improved by adequately modeling cross-channel dependencies. Figure 1 shows two channels which correspond to the occupancy rates in two locations. The presence of cross-channel dependencies is evident through the asynchronism between them, where the peaks in channel 6 lead those in channel

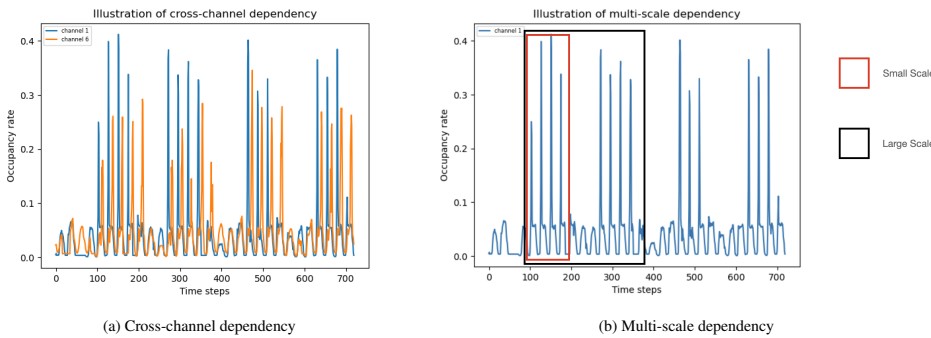

Figure 1: Visualization of Cross-channel dependency and Multi-scale temporal dependency in Traffic dataset

1. Ineffectual modeling of such correlations significantly impacts forecasting results and leads to sub-optimal performance. In this paper, we categorize models with explicit channel dependencies as channel-dependent (CD) models. Conversely, models without channel dependencies are referred to as channel-independent (CI) models.

Another challenge encountered in transformer-based MLTSF models pertains to **fixed-scale awareness**. In this context, the term 'scale' denotes the size of the elementary units when processing a time series. Previous models typically employ a fixed scale for these units (Nie et al., 2022; Wu et al., 2021; Zhou et al., 2022). However, by adhering to a fixed scale, these models fail to capture various dependency patterns inherent within the data. As depicted in Figure 1, larger scales exhibit long-term patterns, while smaller scales reveal short-term patterns within the historical horizon. Thus, we posit that by adequately modeling the temporal dependencies of variables across different scales, it is possible to achieve improved performance. Recent works such as Crossformer (Zhang & Yan, 2022) and Scaleformer (Shabani et al., 2022) explored the potential of multi-scale frameworks for MLTSF. However, the sequential architectures that representations at larger scales are constructed based on the results from smaller scales in these models are susceptible to error accumulation at each scale, resulting in inferior performance when compared to fixed-scale models.

To fill these gaps, we propose MGTST which considers the multi-scale dependency and cross-channel dependency simultaneously. Specifically, MGTST introduces Temporal Embedding with Representation Tokens (TERT), a technique that partitions and projects the original time-series data into temporal embedding tensors. The tensor incorporates appended representation tokens to effectively capture the representation of channels or features within the time series. A temporal attention mechanism is employed to capture temporal dependencies. Subsequently, Cross-Channel Attention and Gated Mechanism (CCAGM) is applied to capture cross-channel dependencies by means of a self-attention mechanism between the representation tokens and dot-product operations between tokens and embedding tensors. Channel Grouping (CG) is included to group the representation tokens, thereby confining the range of interactions and reducing interaction redundancy. Moreover, MGTST incorporates Parallel Multi-Scale Architecture (PMSA), which involves the use of different patch lengths and stride lengths at each scale to generate temporal embedding tensors of varying scales. These tensors are then concatenated and projected to generate the output. **The contributions of this paper can be summarized as follows:**

**1)** We propose MGTST, a transformer-based model that leverages cross-channel dependency learning and multi-scale dependency learning in a unified framework.

**2)** We propose a channel grouping strategy aimed at reducing interaction redundancy. Through this strategy, we demonstrate that conducting channel interactions within a local range yields more effective results compared to interactions across the global range.

**3)** With intensive empirical analysis, we show that our model outperforms both CD and CI models in terms of prediction accuracy, and achieves lower FLOPs and Params in practice compared with current transformer models.

## 2 RELATED WORK

**Transformer-based models for MLTSF.** Recent research efforts have been dedicated to enhancing transformer-based models for MLTSF. Informer (Zhou et al., 2021) reduces the time complexity by introducing ProbSparse self-attention, resulting in a computational complexity of $\mathcal{O}(LlogL)$, where $L$ represents the input length. Autoformer (Wu et al., 2021) replaces traditional dot-product attention with series-wise auto-correlation attention and proposes a seasonal-trend decomposition method based on the temporal characteristics of MLTSF. FEDformer (Zhou et al., 2022) incorporates Fourier analysis into the model, leveraging its properties to further enhance forecasting accuracy. However, these transformer models all adopt the channel embedding strategy which limits their performance. PatchTST (Nie et al., 2022) bridges this gap by introducing the temporal embedding strategy, which transforms independent channel sequences into embedding tensors, achieving state-of-the-art performance. Nevertheless, the exploration of cross-channel dependency and multi-scale frameworks remains relatively limited in the existing works. To incorporate the cross-channel dependency, Crossformer (Zhang & Yan, 2022) utilizes a two-stage attention mechanism to capture both temporal and channel dependencies. CARD (Xue et al., 2023) further tackles the overfitting concerns by integrating a dynamic projection module into the model. However, these models achieve inferior performance compared to PatchTST (Nie et al., 2022), indicating the inadequate modeling of cross-channel dependencies. Targeting the multi-scale frameworks, HUTFormer (Shao et al., 2023) and Crossformer (Zhang & Yan, 2022) both use the sequential architecture to generate multi-scale representations. Scaleformer (Shabani et al., 2022) extends the fixed-scale average pooling to multi-scales. Nevertheless, these multi-scale models exhibit sub-optimal performance when compared to state-of-the-art fixed-scale models. This performance discrepancy can be attributed to the error accumulation in the sequential architecture. To properly model the multi-scale and cross-channel dependencies, MGTST incorporates PMSA and CCAGM to mitigate the error accumulation.

**Non-Transformer based models for MLTSF.** Recurrent neural network(RNN) is one of the major deep learning models used in time-series forecasting tasks. Leveraging their sequential structure, RNN demonstrates the ability to capture the temporal dynamics and causal properties inherent in time-series data. However, the efficacy of RNNs for long time-series forecasting is hindered by the issue of error accumulation, resulting in suboptimal performance (Li & Yang, 2021). Conversely, Multilayer Perceptrons (MLPs) represent another prevalent class of deep learning models for MLTSF. The inherent simplicity of the MLP architecture improves computational efficiency during both model training and inference stages. Notably, several MLP-based models have been proposed, exhibiting competitive performance compared to transformer-based counterparts (Zeng et al., 2023; Das et al., 2023; Li et al., 2023). Despite the effectiveness of MLPs in MLTSF, achieving further performance improvements with such models remains challenging. The most advanced transformer-based models still outperform non-transformer-based models.

**Representation token**. The $[CLS]$ token, originally introduced in Bert (Devlin et al., 2018), has emerged as a symbolic representation token in the field of deep learning. Its incorporation offers notable advantages, such as an initialization with zero bias and the ability to aggregate comprehensive information, thereby rendering it an effective representation of a sentence. This concept has been further extended by Vision Transformer (Dosovitskiy et al., 2020) to capture the representation of images. Inspired by these foundational works, we introduce a representation token to represent each channel in the time series. To the best of our knowledge, this is the first work to introduce the representation token to a transformer-based model in multivariate time series forecasting tasks. By incorporating these tokens, we reduce the computational cost associated with cross-channel interaction and enhance the accuracy of forecasting. This novel idea distinguishes our approach from previous work, such as the study conducted by (Zhang & Yan, 2022). Our empirical results demonstrate the effectiveness of representation tokens in improving forecasting accuracy.

## 3 MODEL ARCHITECTURE

The MLTSF problem can be formulated as follows: given the historical observations of the time series $\boldsymbol{X}_{1:T} \in \mathbb{R}^{T \times M}$, the goal is to predict the future values $\boldsymbol{X}_{T+1:T+\tau} \in \mathbb{R}^{\tau \times M}$, where $T$ is the length of the observations, $\tau$ is the length of the predictions (generally longer than 48), and $M > 1$ is the number of channels. The model architecture of MGTST is depicted in Figure 2. MGTST adopts the **Parallel Multiple-Scale Architecture**. At each scale, it takes $\boldsymbol{X}_{T+1:T+\tau} \in \mathbb{R}^{\tau \times M}$ as

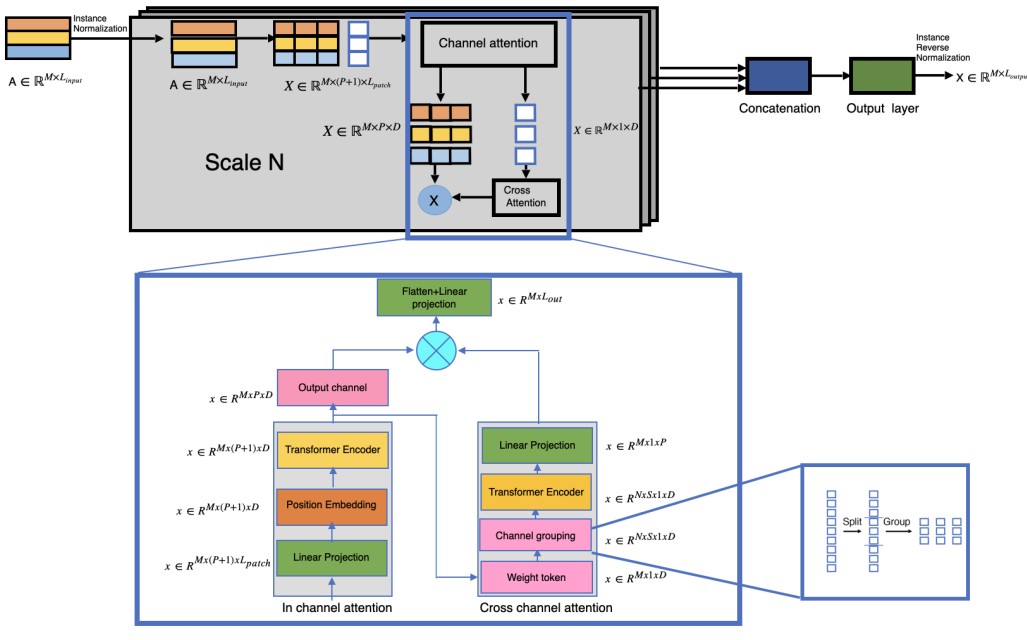

Figure 2: Visualization of model structure

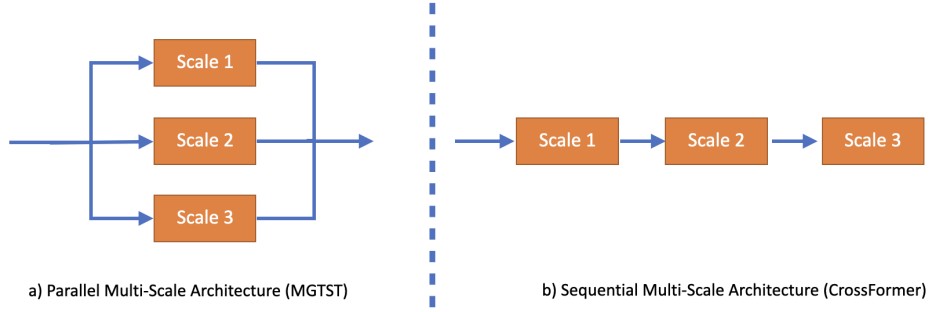

a) Parallel Multi-Scale Architecture (MGTST)

b) Sequential Multi-Scale Architecture (CrossFormer)

Figure 3: Different schemes of Multi-Scale Architecture

the input and transforms the input into the temporal embedding tensor $\boldsymbol{X} \in \mathbb{R}^{M \times (N+1) \times D}$, where $N = \frac{L_{input} - L_{patch}}{S_{stride}} + 1$. This is done through the **Temporal Embedding with Representation Tokens** module. Then the temporal embedding tensor is fed into temporal attention to extract the temporal information. Through the **Cross-Channel Attention and Gated Mechanism** module, representation tokens $\mathbf{T} \in \mathbb{R}^{M \times 1 \times D}$ undergo the self-attention mechanism and dot product operation with temporal embedding tensor to extract cross-channel dependency. For the dataset with many channels, we employ the **Channel Grouping** module to reduce interaction redundancy. Finally, outputs from each scale are concatenated and fed into the predictor for generating prediction $\mathbf{X}^{prediction} \in \mathbb{R}^{M \times L_{output}}$, where $L_{output}$ is the output length.

## 3.1 PARALLEL MULTI-SCALE ARCHITECTURE

Figure 3a) illustrates the Parallel Multi-Scale Architecture in MGTST. A hyperparameter $k$ determines the number of scales. At each scale, the patch length is calculated as $L_{patch} = L_0 * i$ and the stride length is calculated as $S_{stride} = S_0 * i$, where $L_0$ represents the initial patch length, $S_0$ represents the initial stride length, and $i$ represents the index of the scale. The outputs of each scale $\{X_i^{scale}\}$ $(i \in [1, k])$ are concatenated for prediction:

$$\hat{\mathbf{X}} = [\mathbf{X}_0^{scale}, \mathbf{X}_1^{scale}, ..., \mathbf{X}_n^{scale}] \tag{1}$$

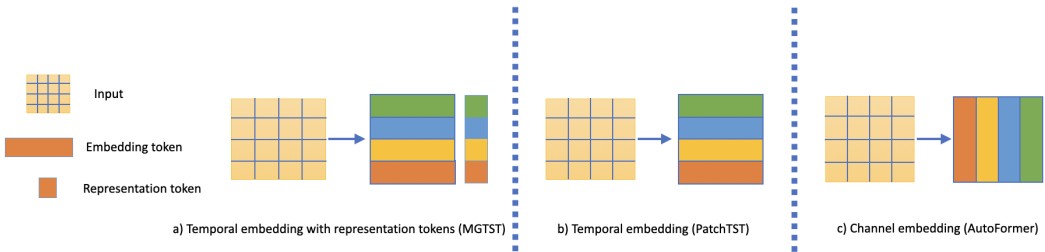

Figure 4: Different schemes of embedding

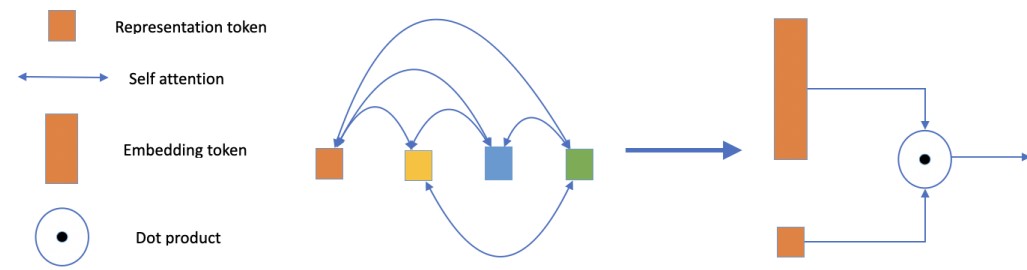

Figure 5: Cross-channel attention and Gated mechanism

The parallel multi-scale architecture used in MGTST (Figure 3a) offers advantages over the sequential multi-scale architecture employed in Crossformer (Figure 3b). With parallel multi-scale architecture, MGTST avoids error accumulation and achieves superior performance.

## 3.2 TEMPORAL EMBEDDING WITH REPRESENTATION TOKENS

Figure 4a) illustrates the Temporal Embedding with Representation Tokens module. Initially, an input tensor $A \in \mathbb{R}^{M \times L_{input}}$ is partitioned into $P$ patches using a moving kernel given the patch length $L_{patch}$ and the stride length $S_{stride}$, where $L_{input}$ is the length of the input. Patches can be represented by $\mathbf{X} \in \mathbb{R}^{M \times N \times L_{patch}}$, where $N = \frac{L_{input} - L_{patch}}{S_{stride}} + 1$. The objective of patching is twofold: capturing local semantic information and reducing the computational complexity associated with the self-attention mechanism. Subsequently, a representation token $\mathbf{T} \in \mathbb{R}^{M \times 1 \times L_{patch}}$ is randomly initialized and concatenated with the patches, increasing the patch number to $N + 1$. In order to project the patches into a temporal embedding space, a linear projection layer is applied, transforming the tensor into $[\mathbf{T}, \mathbf{X}] \in \mathbb{R}^{M \times (N+1) \times D}$, where $D$ signifies the dimensionality of the hidden space. A positional embedding tensor $\mathbf{E} \in \mathbb{R}^{M \times (N+1) \times D}$ is added to the temporal embedding tensor, leading to the temporal tensor representation $\hat{\mathbf{X}} = [\mathbf{T}, \mathbf{X}] + \mathbf{E}$, where $\hat{\mathbf{X}} \in \mathbb{R}^{M \times (N+1) \times D}$.

Compared to the temporal embedding in PatchTST (Figure 4b), MGTST incorporates representation tokens to capture channel-specific information and facilitate cross-channel interaction. Autoformer adopts the channel embedding strategy (Figure 4c) to encode multiple channels into a vector representation, which has been demonstrated to be less effective than the temporal embedding strategy (Nie et al., 2022).

## 3.3 CROSS-CHANNEL ATTENTION AND GATED MECHANISM

Figure 5 illustrates the Cross-Channel Attention and Gated Mechanism module. The cross-channel attention mechanism operates on the representation tokens. Representation token $\mathbf{T} \in \mathbb{R}^{M \times 1 \times D}$ is updated by a standard transformer module:

$$\mathbf{Q} = \boldsymbol{F}_q(\mathbf{T}), \mathbf{K} = \boldsymbol{F}_k(\mathbf{T}), \mathbf{V} = \boldsymbol{F}_v(\mathbf{T}) \tag{2}$$

$$\hat{\mathbf{T}} = BatchNorm(\mathbf{T} + Multihead(\mathbf{Q}, \mathbf{K}, \mathbf{V})) \tag{3}$$

$$\mathbf{T} = (\hat{\mathbf{T}} + MLP(\hat{\mathbf{T}})) \tag{4}$$

where $\mathbf{Q}, \mathbf{K}, \mathbf{V} \in \mathbb{R}^{M \times 1 \times D}$ denote the query, key, and value, and $\boldsymbol{F}_q, \boldsymbol{F}_k, \boldsymbol{F}_v : D \rightarrow D$ are linear projections. $N_{head}$ heads split the $\mathbf{Q}, \mathbf{K}, \mathbf{V}$ as collections of $\mathbf{Q}_i, \mathbf{K}_i, \mathbf{V}_i$, where $\mathbf{Q}_i, \mathbf{K}_i, \mathbf{V}_i \in \mathbb{R}^{M \times 1 \times d_{head}}$, $d_{head} = \frac{D}{N_{head}}$.

Subsequently, the representation token $\mathbf{T}$ undergoes a projection to a dimension of $N$ and is then multiplied with the temporal embedding. This operation yields the adjusted temporal representation for a single scale, denoted as $\mathbf{X}^{scale} \in \mathbb{R}^{M \times N \times D}$:

$$\mathbf{X}^{scale} = Sigmoid(\boldsymbol{L}(\mathbf{T})) * \mathbf{X} \tag{5}$$

where $\boldsymbol{L} : D \rightarrow N$ is a linear projection, $D$ is the hidden dimension, $N$ is the number of patches, and $Sigmoid$ denotes the sigmoid activation function.

### 3.4 CHANNEL GROUPING

For datasets with a large number of channels, Channel Grouping module is employed to mitigate redundancy in channel interaction. The first step involves separating the representation tokens into $G$ groups. Subsequently, we perform cross-channel attention within each group to facilitate the localized modeling of channel interactions:

$$\mathbf{T}_i^{group} = [\mathbf{T}_{i*s}, \mathbf{T}_{i*s+1}...\mathbf{T}_{(i+1)*s}] \tag{6}$$

where $S$ is the size of each group. For the first $G - 1$ group, $S_i = \lfloor \frac{M}{G} \rfloor$. For the last group, $S_{-1} = M - (G - 1) * \lfloor \frac{M}{G} \rfloor$. After local channel interaction, groups are gathered and sent into the gated mechanism module.

## 4 EXPERIMENT

### 4.1 MULTIVARIATE LONG-TERM FORECASTING

**Datasets**. We evaluate MGTST on common datasets (Nie et al., 2022) covering various applications, including Electricity Transformer Temperature (ETTh1, ETTh2, ETTm1, and ETTm2), Weather, Traffic, and Electricity. The details of the datasets are provided in Appendix A.1.1.

**Baselines**. We compare the proposed model with 6 state-of-the-art baselines from both channel-dependent(CD) and channel-independent(CI) categories, including 1) **Crossformer** (Zhang & Yan, 2022)(CD), 2) **Autoformer** (Wu et al., 2021)(CD), 3) **PatchTST** (Nie et al., 2022)(CI), 4) **DLinear** (Zeng et al., 2023)(CI), 5) **TimesNet** (Wu et al., 2022)(CI), and 6) **Client** (Gao et al., 2023)(CD).

**Parameter setting**. We adopt two settings for MGTST in the experiment. The one with the input length of 336 (MGTST-336) is to compare with other models and the one with the input length of 512 (MGTST-512) is to explore the potential of our model. For a fair comparison, we keep the same input length of 336 for all models. The effect of input length is discussed in Section 4.3. More details about hyperparameter settings can be found in the Appendix A.1.4.

**Evaluation Metrics**. For details about Evaluation Metrics, see Appendix A.1.5

**Results**. The prediction results of all models are summarized in Table 1. In general, our model outperforms all CD and CI models. Specifically, MGTST-336 outperforms PatchTST, the best CI model, by 1.5 % on MSE and 1 % on MAE, demonstrating its superior performance and effectiveness of incorporating cross-channel and multi-scale dependencies. It outperforms Client, the current best CD model, by 7.9 % on MSE and 5.3 % on MAE, which indicates both CCAGM and PMSA are more suitable cross-channel mechanism and multi-scale architecture when compared to previous work for MLTSF tasks. Furthermore, MGTST-512 attains a 2.5 % reduction on MSE and 1 % on MAE compared to PatchTST, and attains an 8.8 % reduction on MSE and 5.3 % on MAE compared to Client. The consistency of MGTST across different random seeds can be observed due to its low standard deviation. The code to reproduce our results is available at: https://anonymous.4open.science/r/MGTST-4860

Table 1: Multivariate long sequence time-series forecasting results in seven datasets. **Bold**/underline denotes the best/second result. The Average denotes the average of results for each model (four cases). For each setting, we report the average performance of 4 runs with different seeds with the standard deviation. We use four forecasting window lengths of $h \in \{96, 192, 336, 720\}$ and a look-back window length of $l = 336$ in our experiments.

| Models | | MGTST-336 | | MGTST-512 | | PatchTST | | Crossformer | | DLinear | | Autoformer | | TimesNet | | Client | |
|---|---|---|---|---|---|---|---|---|---|---|---|---|---|---|---|---|---|
| Metric | | MSE | MAE | MSE | MAE | MSE | MAE | MSE | MAE | MSE | MAE | MSE | MAE | MSE | MAE | MSE | MAE |
| ETTh1 | 96 | **0.372** ±0.001 | **0.394** ±0.000 | 0.376 ±0.003 | 0.404 ±0.003 | 0.376 ±0.003 | 0.401 ±0.003 | 0.394 ±0.001 | 0.421 ±0.001 | 0.376 ±0.006 | 0.399 ±0.006 | 0.512 ±0.025 | 0.497 ±0.017 | 0.459 ±0.020 | 0.461 ±0.010 | 0.398 ±0.002 | 0.414 ±0.001 |
| | 192 | 0.418 ±0.002 | 0.422 ±0.002 | **0.403** ±0.001 | 0.421 ±0.001 | 0.412 ±0.002 | **0.420** ±0.001 | 0.426 ±0.001 | 0.442 ±0.001 | 0.418 ±0.018 | 0.427 ±0.016 | 0.493 ±0.017 | 0.495 ±0.012 | 0.477 ±0.009 | 0.471 ±0.004 | 0.450 ±0.001 | 0.450 ±0.001 |
| | 336 | 0.420 ±0.001 | 0.429 ±0.001 | **0.399** ±0.000 | **0.425** ±0.001 | 0.427 ±0.005 | 0.433 ±0.004 | 0.453 ±0.007 | 0.465 ±0.004 | 0.453 ±0.019 | 0.465 ±0.016 | 0.507 ±0.048 | 0.505 ±0.029 | 0.483 ±0.016 | 0.475 ±0.008 | 0.471 ±0.001 | 0.466 ±0.001 |
| | 720 | 0.432 ±0.001 | **0.455** ±0.001 | **0.431** ±0.001 | 0.458 ±0.001 | 0.445 ±0.008 | 0.463 ±0.006 | 0.543 ± 0.080 | 0.541 ±0.051 | 0.478 ±0.009 | 0.494 ±0.008 | 0.597 ±0.030 | 0.567 ±0.022 | 0.534 ±0.025 | 0.512 ±0.011 | 0.492 ±0.001 | 0.496 ±0.001 |
| ETTh2 | 96 | 0.276 ±0.001 | 0.337 ±0.001 | **0.265** ±0.001 | **0.331** ±0.001 | 0.274 ±0.001 | 0.335 ±0.001 | 0.752 ±0.046 | 0.588 ±0.019 | 0.292 ±0.007 | 0.356 ±0.008 | 0.497 ±0.052 | 0.520 ±0.034 | 0.376 ±0.004 | 0.415 ±0.003 | 0.322 ±0.004 | 0.368 ±0.003 |
| | 192 | 0.336 ±0.002 | 0.379 ±0.001 | **0.326** ±0.003 | **0.374** ±0.002 | 0.338 ±0.001 | 0.378 ±0.002 | 0.889 ±0.051 | 0.686 ±0.022 | 0.374 ±0.017 | 0.411 ±0.011 | 0.532 ±0.048 | 0.545 ±0.029 | 0.419 ±0.012 | 0.443 ±0.007 | 0.403 ±0.007 | 0.420 ±0.004 |
| | 336 | 0.326 ±0.001 | 0.383 ±0.002 | **0.323** ±0.002 | 0.381 ±0.001 | 0.328 ±0.002 | **0.380** ±0.002 | 0.944 ±0.101 | 0.723 ±0.054 | 0.432 ±0.036 | 0.450 ±0.019 | 0.692 ±0.174 | 0.616 ±0.065 | 0.399 ±0.009 | 0.436 ±0.006 | 0.441 ±0.020 | 0.450 ±0.011 |
| | 720 | 0.384 ±0.013 | 0.423 ±0.009 | **0.371** ±0.001 | **0.418** ±0.001 | 0.377 ±0.002 | 0.420 ±0.002 | 1.271 ±0.226 | 0.866 ±0.092 | 0.602 ±0.022 | 0.549 ±0.009 | 0.986 ±0.307 | 0.709 ±0.113 | 0.451 ±0.007 | 0.468 ±0.004 | 0.475 ±0.016 | 0.475 ±0.008 |
| ETTm1 | 96 | 0.284 ±0.002 | **0.337** ±0.001 | **0.283** ±0.001 | 0.338 ±0.001 | 0.290 ±0.002 | 0.341 ±0.001 | 0.305 ±0.002 | 0.359 ±0.002 | 0.300 ±0.001 | 0.344 ±0.003 | 0.494 ±0.019 | 0.483 ±0.017 | 0.315 ±0.011 | 0.367 ±0.010 | 0.306 ±0.004 | 0.352 ±0.003 |
| | 192 | **0.322** ±0.001 | **0.363** ±0.001 | 0.325 ±0.001 | 0.365 ±0.001 | 0.331 ±0.004 | 0.368 ±0.002 | 0.357 ±0.004 | 0.399 ±0.004 | 0.335 ±0.001 | 0.366 ±0.002 | 0.535 ±0.043 | 0.502 ±0.021 | 0.375 ±0.012 | 0.397 ±0.005 | 0.341 ±0.003 | 0.368 ±0.002 |
| | 336 | 0.357 ±0.001 | 0.384 ±0.001 | **0.355** ±0.002 | **0.383** ±0.001 | 0.365 ±0.001 | 0.390 ±0.001 | 0.438 ±0.013 | 0.452 ±0.011 | 0.375 ±0.006 | 0.393 ±0.009 | 0.540 ±0.031 | 0.510 ±0.011 | 0.403 ±0.007 | 0.418 ±0.002 | 0.375 ±0.002 | 0.388 ±0.002 |
| | 720 | **0.410** ±0.004 | **0.415** ±0.002 | 0.414 ±0.005 | 0.417 ±0.001 | 0.416 ±0.002 | 0.422 ±0.001 | 0.563 ±0.007 | 0.530 ±0.006 | 0.433 ±0.013 | 0.430 ±0.015 | 0.539 ±0.015 | 0.512 ±0015 | 0.461 ±0.009 | 0.450 ±0.003 | 0.432 ±0.002 | 0.419 ±0.001 |
| ETTm2 | 96 | **0.161** ±0.001 | **0.249** ±0.001 | 0.162 ±0.001 | 0.252 ±0.002 | 0.164 ±0.001 | 0.254 ±0.001 | 0.276 ±0.025 | 0.355 ±0.016 | 0.166 ±0.002 | 0.258 ±0.005 | 0.288 ±0.006 | 0.363 ±0.005 | 0.187 ±0.004 | 0.273 ±0.002 | 0.171 ±0.004 | 0.260 ±0.004 |
| | 192 | **0.217** ±0.001 | **0.289** ±0.001 | 0.217 ±0.001 | 0.289 ±0.001 | 0.220 ±0.001 | 0.292 ±0.001 | 0.436 ±0.040 | 0.490 ±0.027 | 0.229 ±0.006 | 0.306 ±0.006 | 0.345 ±0.021 | 0.395 ±0.019 | 0.250 ±0.008 | 0.319 ±0.004 | 0.227 ±0.010 | 0.297 ±0.005 |
| | 336 | 0.273 ±0.003 | 0.325 ±0.002 | **0.266** ±0.001 | **0.321** ±0.001 | 0.275 ±0.002 | 0.328 ±0.002 | 0.798 ±0.063 | 0.642 ±0.039 | 0.296 ±0.009 | 0.357 ±0.009 | 0.467 ±0.053 | 0.468 ±0.024 | 0.299 ±0.007 | 0.348 ±0.006 | 0.285 ±0.011 | 0.334 ±0.006 |
| | 720 | 0.360 ±0.004 | 0.380 ±0.002 | **0.349** ±0.003 | **0.377** ±0.003 | 0.364 ±0.002 | 0.382 ±0.001 | 1.760 ±0.139 | 1.018 ±0.060 | 0.424 ±0.025 | 0.433 ±0.015 | 0.492 ±0.072 | 0.462 ±0.026 | 0.393 ±0.010 | 0.403 ±0.006 | 0.378 ±0.004 | 0.393 ±0.002 |
| Electricity | 96 | 0.127 ±0.001 | **0.221** ±0.001 | **0.127** ±0.000 | 0.221 ±0.001 | 0.129 ±0.001 | 0.222 ±0.001 | 0.146 ±0.001 | 0.251 ±0.001 | 0.139 ±0.001 | 0.237 ±0.001 | 0.206 ±0.011 | 0.320 ±0.010 | 0.180 ±0.003 | 0.286 ±0.002 | 0.132 ±0.001 | 0.227 ±0.001 |
| | 192 | 0.146 ±0.001 | **0.238** ±0.001 | **0.146** ±0.000 | 0.239 ±0.001 | 0.148 ±0.001 | 0.240 ±0.001 | 0.167 ±0.005 | 0.270 ±0.005 | 0.152 ±0.000 | 0.249 ±0.000 | 0.217 ±0.007 | 0.331 ±0.006 | 0.230 ±0.023 | 0.322 ±0.016 | 0.153 ±0.001 | 0.247 ±0.002 |
| | 336 | 0.163 ±0.001 | **0.256** ±0.001 | **0.161** ±0.001 | 0.256 ±0.000 | 0.164 ±0.001 | 0.258 ±0.001 | 0.192 ±0.004 | 0.294 ±0.003 | 0.168 ±0.000 | 0.267 ±0.000 | 0.223 ±0.004 | 0.336 ±0.003 | 0.238 ±0.023 | 0.329 ±0.015 | 0.169 ±0.001 | 0.265 ±0.001 |
| | 720 | 0.200 ±0.001 | 0.289 ±0.001 | **0.196** ±0.001 | **0.288** ±0.001 | 0.203 ±0.004 | 0.291 ±0.001 | 0.258 ±0.003 | 0.349 ±0.003 | 0.202 ±0.000 | 0.300 ±0.000 | 0.253 ±0.015 | 0.355 ±0.010 | 0.285 ±0.005 | 0.364 ±0.007 | 0.210 ±0.002 | 0.301 ±0.002 |
| Weather | 96 | 0.144 ±0.001 | 0.193 ±0.001 | **0.142** ±0.001 | **0.192** ±0.001 | 0.150 ±0.001 | 0.198 ±0.001 | 0.147 ±0.001 | 0.212 ±0.001 | 0.175 ±0.001 | 0.237 ±0.005 | 0.297 ±0.015 | 0.373 ±0.015 | 0.186 ±0.007 | 0.245 ±0.006 | 0.165 ±0.001 | 0.216 ±0.001 |
| | 192 | 0.187 ±0.001 | 0.235 ±0.001 | **0.187** ±0.002 | **0.235** ±0.002 | 0.195 ±0.001 | 0.241 ±0.001 | 0.194 ±0.001 | 0.261 ±0.001 | 0.215 ±0.001 | 0.274 ±0.001 | 0.390 ±0.038 | 0.438 ±0.030 | 0.233 ±0.011 | 0.280 ±0.007 | 0.208 ±0.004 | 0.255 ±0.004 |
| | 336 | 0.238 ±0.001 | 0.275 ±0.002 | **0.236** ±0.001 | **0.274** ±0.001 | 0.247 ±0.001 | 0.282 ±0.001 | 0.245 ±0.001 | 0.306 ±0.003 | 0.261 ±0.001 | 0.312 ±0.003 | 0.425 ±0.031 | 0.450 ±0.025 | 0.277 ±0.012 | 0.307 ±0.007 | 0.254 ±0.003 | 0.290 ±0.002 |
| | 720 | 0.310 ±0.001 | 0.329 ±0.003 | **0.304** ±0.001 | **0.326** ±0.001 | 0.317 ±0.001 | 0.333 ±0.001 | 0.319 ±0.005 | 0.360 ±0.004 | 0.325 ±0.001 | 0.366 ±0.006 | 0.513 ±0.061 | 0.485 ±0.033 | 0.342 ±0.003 | 0.351 ±0.002 | 0.324 ±0.002 | 0.336 ±0.002 |
| traffic | 96 | **0.361** ±0.001 | 0.250 ±0.001 | 0.364 ±0.001 | 0.253 ±0.000 | 0.366 ±0.001 | **0.249** ±0.001 | 0.496 ±0.001 | 0.280 ±0.001 | 0.410 ±0.000 | 0.281 ±0.000 | 0.667 ±0.025 | 0.406 ±0.019 | 0.599 ±0.004 | 0.327 ±0.002 | 0.365 ±0.002 | 0.264 ±0.002 |
| | 192 | 0.384 ±0.002 | 0.260 ±0.001 | **0.380** ±0.001 | 0.261 ±0.001 | 0.385 ±0.002 | **0.259** ±0.003 | 0.513 ±0.004 | 0.288 ±0.002 | 0.421 ±0.000 | 0.286 ±0.000 | 0.663 ±0.011 | 0.405 ±0.010 | 0.617 ±0.007 | 0.337 ±0.006 | 0.390 ±0.001 | 0.275 ±0.001 |
| | 336 | 0.400 ±0.003 | 0.269 ±0.003 | **0.391** ±0.002 | 0.268 ±0.001 | 0.398 ±0.001 | **0.265** ±0.001 | 0.538 ±0.004 | 0.300 ±0.003 | 0.435 ±0.000 | 0.295 ±0.000 | 0.644 ±0.026 | 0.391 ±0.015 | 0.628 ±0.005 | 0.343 ±0.004 | 0.407 ±0.003 | 0.285 ±0.003 |
| | 720 | **0.436** ±0.005 | 0.290 ±0.005 | 0.437 ±0.002 | 0.294 ±0.002 | 0.440 ±0.011 | 0.293 ±0.012 | 0.747 ±0.002 | 0.409 ±0.003 | 0.465 ±0.000 | 0.314 ±0.000 | 0.652 ±0.009 | 0.401 ±0.010 | 0.729 ±0.091 | 0.391 ±0.045 | 0.442 ±0.002 | 0.303 ±0.004 |
| Average | | 0.302 | 0.324 | **0.298** | **0.324** | 0.306 | 0.327 | 0.520 | 0.448 | 0.334 | 0.352 | 0.488 | 0.459 | 0.387 | 0.376 | 0.328 | 0.343 |

## 4.2 ABLATION STUDY

In this section, we undertake an empirical investigation to assess **the effects of PMSA and CCAGM** in the context of MGTST. To isolate the influence of PMSA, we conduct an ablation experiment by setting the scale number to 1, thereby removing the multi-scale functionality. Similarly, to evaluate the impact of CCAGM, we perform another ablation experiment where we exclude CCAGM. The results of these ablation experiments are presented in Table 2. Upon careful analysis of the findings, the following key observations emerge: **1)** The inclusion of both PMSA and CCAGM yields significant performance improvements, this highlighting the significance of capturing multi-scale dependencies and cross-channel dependencies to the forecasting performance. **2)** The combination of PMSA and CCAGM demonstrates a synergistic effect, resulting in further performance gains compared to the utilization of either mechanism in isolation.

## 4.3 SENSITIVITY STUDY

In this section, we conduct a comprehensive study to investigate the impact of multiple hyperparameters on MGTST. We analyze the effects of **channel group sizes, input lengths, scale numbers, and stride lengths**. The experiment on channel group sizes is performed on the Traffic dataset, given its largest number of channels, while the experiment on input lengths and scale numbers is conducted on the ETTm1 dataset. Regarding **input lengths** (Figure 6a), we observe a positive correlation between input length and forecasting accuracy. However, improvements in accuracy are not significant beyond an input length of 512. For **scale numbers** (Figure 6b), manipulating the

Table 2: Ablation study on PMSA and CCAGM with MGTST. We use four forecasting window lengths $L \in 96, 192, 336, 720$ for all benchmarks with a look-back window length of 336. The best prediction results are in **bold** and the second best are in underline. The Average denotes the average result on all datasets for each model.

| Models | | MGTST | | | |
|---|---|---|---|---|---|
| | | - CCAGM | - PMSA | - both | original |
| Metric | | MSE MAE | MSE MAE | MSE MAE | MSE MAE |
| ETTh1 | 96 | **0.361 0.388** | 0.367 0.394 | 0.366 0.393 | 0.370 0.393 |
| | 192 | **0.406 0.414** | 0.409 0.418 | 0.410 0.420 | 0.419 0.423 |
| | 336 | 0.429 0.438 | 0.423 0.427 | 0.429 0.436 | **0.420 0.429** |
| | 720 | 0.433 0.455 | 0.437 0.465 | 0.462 0.473 | **0.431 0.455** |
| ETTh2 | 96 | 0.280 0.337 | 0.275 0.338 | 0.276 0.336 | **0.275 0.335** |
| | 192 | 0.353 0.386 | **0.335 0.379** | 0.344 0.383 | 0.339 0.381 |
| | 336 | 0.336 **0.381** | **0.326** 0.384 | 0.335 0.384 | 0.327 0.383 |
| | 720 | 0.383 0.422 | **0.378 0.422** | 0.379 0.422 | **0.377 0.419** |
| ETTm1 | 96 | 0.286 0.338 | 0.291 0.342 | 0.299 0.348 | **0.284 0.336** |
| | 192 | 0.325 0.364 | 0.331 0.366 | 0.338 0.370 | **0.325 0.364** |
| | 336 | **0.361 0.385** | 0.365 .387 | 0.376 0.393 | 0.358 0.384 |
| | 720 | 0.408 0.414 | 0.420 0.418 | 0.429 0.422 | **0.408 0.414** |
| ETTm2 | 96 | 0.165 0.254 | 0.164 0.254 | 0.166 0.255 | **0.163 0.251** |
| | 192 | 0.220 0.291 | 0.221 0.294 | 0.224 0.297 | **0.217 0.289** |
| | 336 | 0.274 0.326 | 0.277 0.332 | 0.283 0.336 | **0.273 0.325** |
| | 720 | 0.360 0.380 | 0.370 0.389 | 0.360 0.385 | **0.350 0.378** |
| Electricity | 96 | 0.135 0.229 | 0.133 0.228 | 0.139 0.234 | **0.128 0.221** |
| | 192 | 0.150 0.243 | 0.149 0.242 | 0.153 0.246 | **0.147 0.238** |
| | 336 | 0.166 0.260 | 0.165 0.259 | 0.168 0.263 | **0.164 0.256** |
| | 720 | 0.204 0.292 | 0.203 0.291 | 0.207 0.296 | **0.199 0.288** |
| Weather | 96 | 0.157 0.204 | 0.152 0.203 | 0.172 0.221 | **0.143 0.193** |
| | 192 | 0.200 0.245 | 0.196 0.244 | 0.214 0.256 | **0.187 0.236** |
| | 336 | 0.249 0.282 | 0.246 0.282 | 0.260 0.291 | **0.238 0.278** |
| | 720 | 0.318 0.331 | 0.317 0.334 | 0.328 0.340 | **0.314 0.333** |
| traffic | 96 | 0.375 0.256 | 0.390 0.266 | 0.390 0.268 | **0.371 0.255** |
| | 192 | 0.392 0.263 | 0.404 0.271 | 0.404 0.271 | **0.387 0.262** |
| | 336 | 0.405 0.269 | 0.416 0.278 | 0.424 0.289 | **0.400 0.268** |
| | 720 | 0.437 0.290 | 0.444 0.295 | 0.447 0.300 | **0.430 0.286** |
| Average | | 0.306 0.326 | 0.307 0.328 | 0.313 0.333 | **0.301 0.324** |

scale number does not significantly affect prediction accuracy when the forecasting horizon is 96. However, as the horizon lengthens, an increase in scale number consistently reduces the MSE until reaching a saturation point at a scale number of 3. The experiment on **channel group size** (Figure 6c) shows that increasing the group size decreases MSE, indicating improved performance through limited channel interaction. The optimal group size is approximately 30, beyond which further increases in group size lead to performance decline. Lastly, Figure 6d illustrates that increasing **stride length** increases MSE, particularly for longer forecasting lengths.

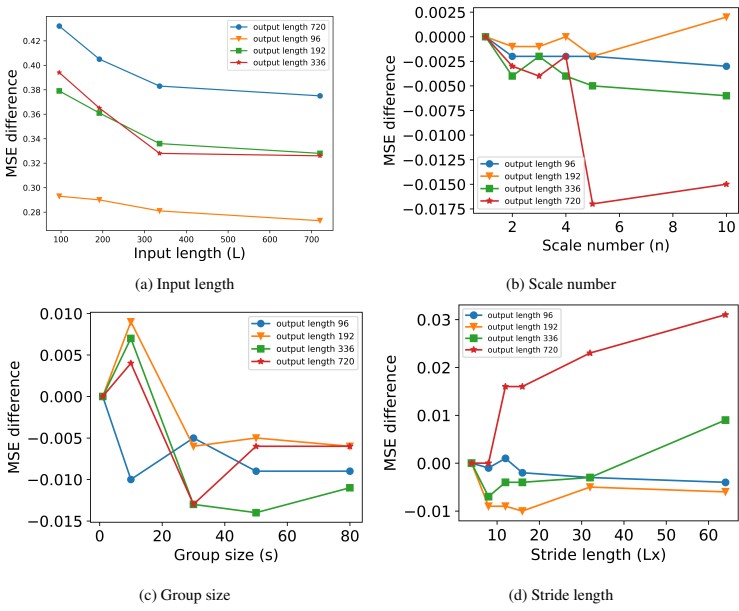

(a) Input length

(b) Scale number

(c) Group size

(d) Stride length

Figure 6: Sensitivety analysis of the four hyperparameters in MGTST.

Table 3: Computational complexity of transformer-based model per layer. $T$ indicates the length of input. $D$ indicates the number of channels. $L_{seq}$ indicates the stride length of patch.

| Models | Encoder layer | decoder layer |
|---|---|---|
| Transformer (Vaswani et al., 2017) | $O(T^2)$ | $O(\tau(\tau + T))$ |
| Informer (Zhou et al., 2021) | $O(T \log T)$ | $O(\tau(\tau + \log T))$ |
| FEDformer (Zhou et al., 2022) | $O(T)$ | $O(\frac{T}{2} + \tau))$ |
| Autoformer (Wu et al., 2021) | $O(T \log T)$ | $O((\frac{T}{2} + \tau) \log(\frac{T}{2} + \tau)))$ |
| Crossformer (Zhang & Yan, 2022) | $O(\frac{D}{L_{seq}^2}T^2)$ | $O(\frac{D}{L_{seq}}\tau(\tau + T))$ |
| PatchTST (Nie et al., 2022) | $O(\frac{D}{L_{seq}^2}T^2)$ | |
| MGTST (ours) | $O(\frac{D}{L_{seq}^2}T^2 + D^2)$ | |

Table 4: FLOPs and Params for each model in default setting on weather dataset.

| Models | | MGTST | PatchTST | Crossformer | DLinear | Autoformer | TimesNet | Client |
|---|---|---|---|---|---|---|---|---|
| Metrics | | FLOPs Params | FLOPs Params | FLOPs Params | FLOPs Params | FLOPs Params | FLOPs Params | FLOPs Params |
| Weather | 96 | 3.48 0.38 | 46.35 0.91 | 157.03 11.07 | 0.17 1.35 | 363.79 10.60 | 330.46 1.32 | 2.73 1.01 |
| | 192 | 4.26 0.68 | 47.74 1.43 | 109.64 11.08 | 0.347 2.71 | 409.86 10.60 | 397.25 1.35 | 2.90 1.08 |
| | 336 | 5.44 1.11 | 49.82 2.20 | 78.75 11.10 | 0.60 4.75 | 478.96 10.60 | 515.85 1.40 | 3.16 1.17 |
| | 720 | 8.58 2.28 | 55.37 4.27 | 141.02 11.10 | 1.31 10.31 | 663.22 10.60 | 794.13 1.53 | 3.86 1.43 |

## 4.4 COMPUTATIONAL EFFICIENCY ANALYSIS

Theoretical complexity analysis per layer is conducted on typical transformer-based time-series models, and the results are presented in Table 3. Both MGTST and PatchTST (Nie et al., 2022) are encoder-only models, rendering the decoder complexity omittable. The complexity can be reduced by increasing the stride length, denoted as $L_{seq}$. However, the performance of the models also varied across different stride lengths according to Section 4.3. Thus, enhancing speed by augmenting the stride length is impractical due to the detrimental impact on accuracy.

Furthermore, we compare the running time for different models with the default settings corresponding to the results in Table 1. The batch size for each model is modified to 128, as it has a minimal impact on forecasting accuracy but exerts a substantial influence on computational complexity. Two complexity metrics, namely floating point operations per second (FLOPs) and the number of parameters (Params), are employed for evaluation. The observed result is shown in Table 4. Notably, the FLOPs of MGTST rank as the third lowest among all models, while the Params associated with MGTST are the lowest. The reduction of model parameters in MGTST is accomplished by diminishing the dimension of the latent space and the depth of the model.

The assessment also considers the impact of different components on the complexity. We leave the detailed discussion in Section A.5. In summary, the multi-scale architecture is found to contribute the most to the complexity.

## 5 CONCLUSION AND FUTURE WORK

In light of the significance of multivariate long-term forecasting, we introduce MGTST, a novel transformer-based model that incorporates PMSA and CCAGM to capture the temporal dependency across different scales and cross-channel dependencies effectively. Furthermore, we propose a CG strategy that reduces channel interaction redundancy and enhances overall performance. Through empirical evaluation, we demonstrate the superiority of our model over six state-of-the-art models, including both channel-dependent and channel-independent models, achieving an average improvement in mean squared error (MSE) ranging from 1.5 percent to 41.9 percent. Nevertheless, the multi-scale strategy applied in the proposed framework is resource-consuming compared to other components since the number of patches increases along with the number of scales. A potential approach to mitigate this issue is to handle each scale in a distributed manner. Therefore, embedding tensors with different scale can be processed in parallel.

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

# A APPENDIX

## A.1 EXPERIMENT DETAILS

### A.1.1 DATA DESCRIPTION

The detailed descriptions of the data are as follows: 1) ETT (Electricity Transformer Temperature) contains 2-year data from two separate Chinese countries (Zhou et al., 2021). ETTh indicates the ETT data with a granularity of 1-hour-level and ETTm indicates the ETT data with a granularity of 15-minutes-level. The raw data contains seven features including the oil temperature and six power load features. 2) Electricity contains hourly electricity consumption from 321 customers (Trindade, 2015). 3) Weather dataset contains 21 meteorological indicators in Germany (jena, 2020). 4) Traffic contains the road occupancy rates from 862 sensors on San Francisco freeways. The channel numbers and timesteps of each dataset are described in Table 5. For the ETT dataset, we divide the raw data into the training/validation/testing parts following a ratio of 0.6/0.2/0.2. For the others, we apply a ratio of 0.7/0.1/0.2 to remain consistent with previous works.

Table 5: Channel numbers and timesteps of each dataset.

| Datasets | Weather | Traffic | Electricity | ETTh1 | ETTh2 | ETTm1 | ETTm2 |
|---|---|---|---|---|---|---|---|
| Channel numbers | 21 | 862 | 321 | 7 | 7 | 7 | 7 |
| Timesteps | 52696 | 17544 | 26304 | 17420 | 17420 | 69680 | 69680 |

### A.1.2 EVALUATION METRICS

### A.1.3 BASELINE DETAILS

In this section, we summarize the state-of-the-art models that have been compared with MGTST in previous experiments.

1) PatchTST (Nie et al., 2022) groups time-series points into patches. Then it applies patch-wise attention with Feed Forward Layer to extract information and make predictions. It considers each channel separately.

2) DLinear (Zeng et al., 2023) decomposes the input into seasonal and trend parts. Then both parts are processed by a single linear layer and summed up in the output.

3) Crossformer (Zhang & Yan, 2022) adds a cross-attention layer on top of PatchTST. It utilizes the router mechanism to improve efficiency and performance.

4) Autoformer (Wu et al., 2021) decomposes the input into seasonal and trend parts. Then it applies attention based on the auto-correlation mechanism which discovers period-based dependency.

5) TimesNet (Wu et al., 2022) extends the analysis of temporal variations into 2D space based on multiple periods to discover the multi-periodicity.

6) Client (Gao et al., 2023) combines the linear model with the transformer model. The linear model is used to model the relationship between different time points while the transformer model is used to model the relationship between different channels.

### A.1.4 HYPER-PARAMETER CHOICE AND IMPLEMENTATION DETAILS

In the main experiment, MGTST contains 3 encoder layers for ETT datasets (ETTh1, ETTh2, ETTm1, ETTm2) and 1 encoder layer for other datasets. MGTST uses 4 heads attention for weather and 8 heads for other datasets. For ETTh1, we set the dimension of latent space D=128 and the dimension of feedforward layer F=256. For ETTh2, traffic and Electricity, D = 64 and F=128. For ETTm1, D=16 and F=64. For ETTm2, D=64 and F=64. For weather, D=32 and F=64. We set the patch length to 16, the stride length to 8, and the dropout rate to 0.3 for all datasets. Group size G = 107 for Electricity and G=30 for Traffic. For other datasets, G=1.

In order to maintain consistency with the original paper, the hyperparameters for the other baseline models are kept the same, except for the input length which was set to 336. This is motivated by

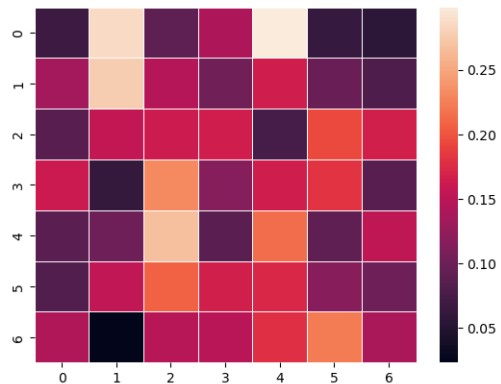

Figure 7: A cross-channel attention map.

the aim to ensure a fair and unbiased comparison, as depicted in Figure 6a, where it is demonstrated that altering the input length has a discernible effect on model performance.

All models are implemented in PyTorch and trained on a NVIDIA A100 GPU with 80GB memory.

### A.1.5 EVALUATION METRICS

We use Mean Square Error (MSE) and Mean Absolute error (MAE) for model evaluation:

$$MAE = \frac{1}{L} \sum_{k=0}^{L} |x_{t+k} - \hat{x}_{t+k}| \tag{7}$$

$$MSE = \frac{1}{L} \sum_{k=0}^{L} |x_{t+k} - \hat{x}_{t+k}|^2 \tag{8}$$

where $L$ is the prediction length, $x_{t+k}$ is the ground truth, and $\hat{x}_{t+k}$ is the prediction result.

### A.2 VISUALIZATION

In this section, a collection of visualizations encompassing attention maps, temporal forecasting results, and frequency patterns of forecasting results are presented to provide elucidation on the discernible dissimilarities among the models. We apply the ETTh1 dataset for all comparisons, with the input length consistently set to 336 and the output length fixed at 96. It is important to note that the parameter configuration adheres to the default settings prescribed by the model. For specification, see A.1.4. The inclusion of these visualizations serves to show the distinctive characteristics exhibited by different models and facilitate a more comprehensive understanding of their respective performances.

### A.2.1 ATTENTION MAP

Figure 7 presents a cross-channel attention map and Figure 8 reveals distinct patterns in the attention maps at varying scales. These divergent patterns suggest that MGTST effectively captures and incorporates information from different scales and different channels.

### A.2.2 TEMPORAL PREDICTION

Figure 9 illustrates the superior forecasting performance of our model over other baseline models. Notably, MGTST exhibits comparable accuracy in predicting the ground truth values. This visualization serves as evidence of our model's effectiveness and its ability to generate highly accurate forecasts when compared to the alternative approaches.

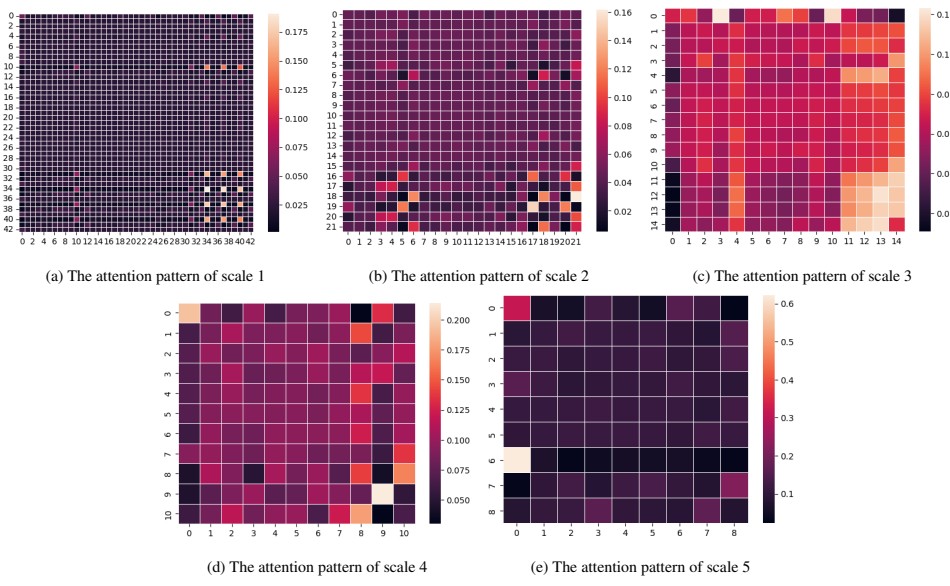

(a) The attention pattern of scale 1  (b) The attention pattern of scale 2  (c) The attention pattern of scale 3

(d) The attention pattern of scale 4  (e) The attention pattern of scale 5

Figure 8: Attention maps of MGTST with 5 scales.

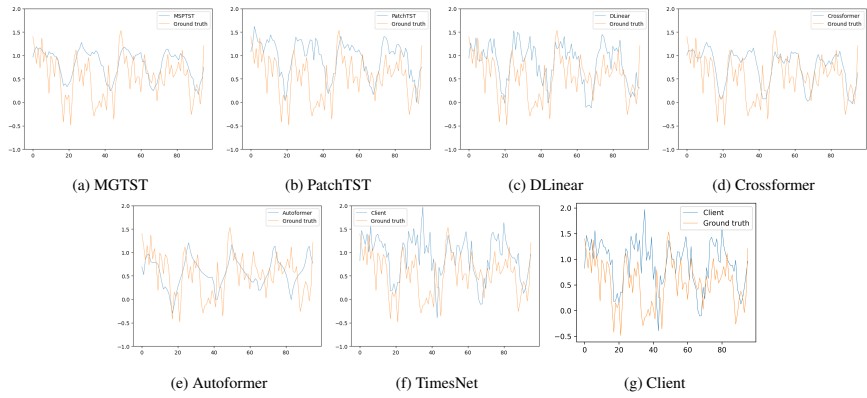

(a) MGTST  (b) PatchTST  (c) DLinear  (d) Crossformer

(e) Autoformer  (f) TimesNet  (g) Client

Figure 9: Forecasting visualization of different models.

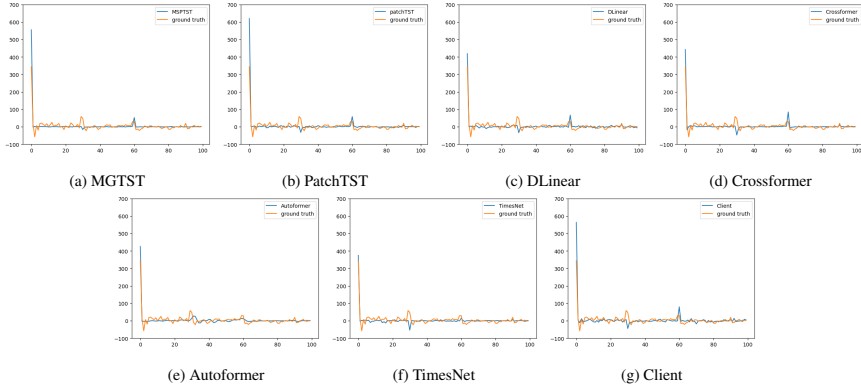

(a) MGTST  (b) PatchTST  (c) DLinear  (d) Crossformer

(e) Autoformer  (f) TimesNet  (g) Client

Figure 10: Forecasting visualization of different models in low-frequency domain.

### A.2.3  FREQUENCY PATTERN

To fully understand the model difference from other aspects, a frequency pattern analysis is conducted. To enhance clarity, the frequency pattern is divided into two segments: the low-frequency

domain (ranging from 0 to 100) and the high-frequency domain (ranging from 100 to 361). Figure 10 illustrates that the MGTST model produces a reasonably accurate prediction of the mean value and successfully captures the third-highest peak in the low-frequency range. Conversely, Figure 11 demonstrates that the forecasting results of the MGTST model exhibit a flat trend in the high-frequency domain. It is intuitive to infer that high-frequency components represent noise in the time series and should be avoided when making forecasts.

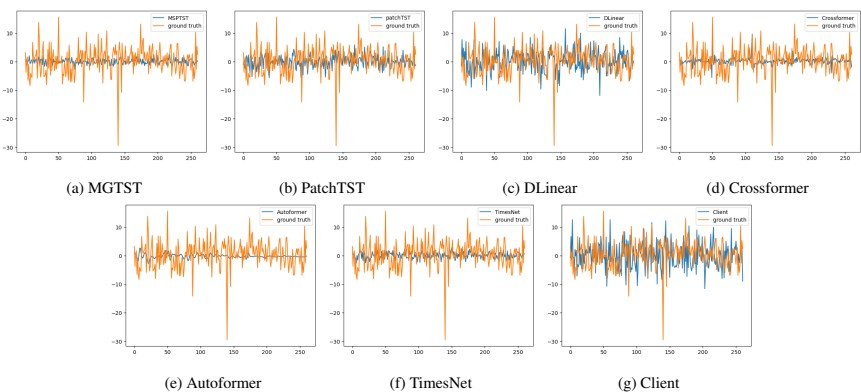

| (a) MGTST | (b) PatchTST | (c) DLinear | (d) Crossformer |
| (e) Autoformer | (f) TimesNet | (g) Client |

Figure 11: Forecasting visualization of different models in high-frequency domain.

### A.3 WORKFLOW

We summarize the workflow as pseudo-code in Algorithm 1, where $patchEmbedding$ denotes **patching**, $TemporalAttention$ denotes **in channel attention**, $Grouping$ denotes **channel grouping**, $SpatialAttention$ denotes **cross-channel attention**, and $FlattenHead$ denotes **multi-Scale architecture** in 2. $S$, $C$, and $G$ denote hyperparameters of scale numbers, channel numbers, and group numbers.

---

**Algorithm 1** MGTST's workflow

---

**Input** : The input MTS history $\mathbf{X}$, the initialized representation token $\mathbf{T}$
**Output** : The output MTS forecasting $\mathbf{Y}$

1: $X^{time}, T^{space} \leftarrow PatchEmbedding(\hat{X}, T)$ ▷ patch embedding
2: **for** $s = 1, 2, 3 \cdots S$ **do**
3:     **for** $c = 1, 2, 3 \cdots C$ **do**
4:         $\hat{\mathbf{X}}_{c,s}^{time} \leftarrow TemporalAttention(\mathbf{X}_{c,s}^{time})$ ▷ applying temporal attention
5:     **end for**
6:     $\mathbf{T}_{s,g}^{space} \leftarrow Grouping(\mathbf{T}_s^{space})$
7:     **for** $g = 1, 2, 3 \cdots G$ **do**
8:         $\hat{\mathbf{T}}_{s,g}^{space} \leftarrow SpatialAttention(\mathbf{T}_{s,g}^{space})$ ▷ applying spatial attention
9:     **end for**
10:    $\hat{\mathbf{T}}_s^{space} \leftarrow Concat(\hat{\mathbf{T}}_{s,g}^{space})$
11:    $\mathbf{T}_s^{weight} \leftarrow LinearProjection(\hat{\mathbf{T}}_s^{space})$
12:    $\mathbf{X}_s^{scale} \leftarrow Sigmoid(\mathbf{T}_s^{weight}) \bullet \hat{\mathbf{X}}_{c,s}^{time}$ ▷ dot product
13: **end for**
14: $\mathbf{X}^{scale} \leftarrow Concat(\mathbf{X}_s^{scale})$
15: return $\mathbf{Y} \leftarrow FlattenHead(\mathbf{X}^{scale})$

---

Table 6: Effects of random seed on MGTST.

| Random seeds | | 1 | 42 | 2021 | 3407 |
|---|---|---|---|---|---|
| Metric | | MSE MAE | MSE MAE | MSE MAE | MSE MAE |
| ETTh1 | 96 | 0.371 0.394 | 0.372 0.394 | 0.373,0.394 | 0.372 0.394 |
| | 192 | 0.415 0.420 | 0.420 0.424 | 0.419 0.424 | 0.419 0.423 |
| | 336 | 0.421 0.430 | 0.421 0.430 | 0.420 0.428 | 0.420 0.429 |
| | 720 | 0.432 0.457 | 0.434 0.456 | 0.431 0.455 | 0.432 0.455 |

Table 7: Sensitivity study on reversible instance normalization.

| revin | | original | - revin | improvement |
|---|---|---|---|---|
| Metric | | MSE MAE | MSE MAE | MSE MAE |
| ETTh1 | 96 | 0.370 0.393 | 0.398 0.411 | -7.0% -4.3% |
| | 192 | 0.419 0.423 | 0.446 0.441 | -6.0% -4.0% |
| | 336 | 0.420 0.429 | 0.459 0.449 | -8.4% -4.4% |
| | 720 | 0.431 0.455 | 0.516 0.494 | -16.4% -7.8% |
| ETTh2 | 96 | 0.275 0.335 | 0.590 0.526 | -53.3% -36.3% |
| | 192 | 0.339 0.381 | 0.700 0.581 | -51.5% -34.4% |
| | 336 | 0.327 0.383 | 0.901 0.664 | -63.7% -42.3% |
| | 720 | 0.377 0.419 | 1.192 0.751 | -68.3% -44.2% |
| ETTm1 | 96 | 0.284 0.336 | 0.288 0.346 | -1.3% -2.8% |
| | 192 | 0.325 0.364 | 0.329 0.373 | -1.2% -2.4% |
| | 336 | 0.358 0.384 | 0.367 0.399 | -2.4% -3.7% |
| | 720 | 0.408 0.414 | 0.427 0.437 | -4.4% -5.2% |
| ETTm2 | 96 | 0.163 0.251 | 0.193 0.283 | -15.5% -11.3% |
| | 192 | 0.217 0.289 | 0.296 0.377 | -26.6% -23.3% |
| | 336 | 0.273 0.325 | 0.542 0.492 | -49.6% -33.9% |
| | 720 | 0.350 0.378 | 1.736 0.922 | -79.8% -59.0% |
| Electricity | 96 | 0.128 0.221 | 0.132 0.229 | -3.0% -3.4% |
| | 192 | 0.147 0.238 | 0.152 0.250 | -3.2% -4.8% |
| | 336 | 0.164 0.256 | 0.167 0.266 | -1.7% -3.7% |
| | 720 | 0.199 0.288 | 0.216 0.306 | -7.8% -5.8% |
| weather | 96 | 0.143 0.193 | 0.147 0.206 | -2.7% -6.3% |
| | 192 | 0.187 0.236 | 0.190 0.247 | -1.5% -4.4% |
| | 336 | 0.238 0.278 | 0.242 0.290 | -1.6% -4.1% |
| | 720 | 0.314 0.333 | 0.313 0.351 | +0.3% -5.1% |
| traffic | 96 | 0.371 0.255 | 0.464 0.267 | -20.0% -4.4% |
| | 192 | 0.387 0.262 | 0.506 0.277 | -23.5% -5.4% |
| | 336 | 0.400 0.268 | 0.511 0.296 | -21.7% -9.4% |
| | 720 | 0.430 0.286 | 0.568 0.312 | -24.2% -8.3% |

## A.4 ADDITIONAL PARAMETER SENSITIVITY

### A.4.1 RANDOM SEEDS

In this section, an analysis is conducted to evaluate the impact of multiple random seeds on the ETTh1 dataset. Specifically, random seeds of 2021, 3407, 1, and 42 are tested. The results, as presented in Table 6, indicate that random seeds have a negligible effect on the performance of MGTST. The observed variance in MSE is approximately 0.65 percent, while the variance in MAE is approximately 0.42 percent. These findings suggest that the choice of random seed does not significantly influence the overall performance of the model.

### A.4.2 INSTANCE NORMALIZATION

In this section, we conduct a comparative analysis between the results obtained with and without instance normalization. The findings are summarized in Table 7, which demonstrates that the incorporation of instance normalization leads to a substantial reduction in both MSE and MAE. Notably, as the length of the output increases, the observed improvement in performance becomes more evident. These results underscore the efficacy of instance normalization in improving the accuracy of forecasting.

## A.5 COMPLEXITY ANALYSIS

We evaluate FLOPs for three variations of MGTST: MGTST without the gate mechanism, MGTST without the multi-scale architecture, and MGTST without both, in relation to the hidden dimension $D$. These variations are evaluated using the weather dataset. As shown in Figure 12, the multi-scale framework significantly increases the complexity, while the gate mechanism only has a minor

impact on complexity. Additionally, it is worth noting that the trend of FLOPs with respect to the hidden dimension $D$ follows a quadratic pattern, aligning with the complexity analysis.

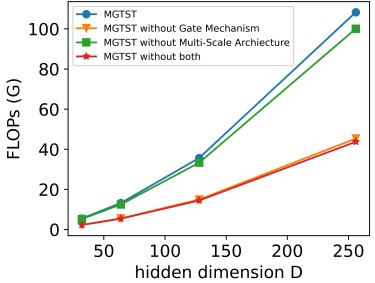

Figure 12: FLOPs of MGTST with different components

## A.6   ALTERNATIVE APPROACH FOR MULTI-SCALE ARCHITECTURE

In this section, we propose an alternative architectural framework for simulating multi-scale dependency that incorporates Multiple Multi-head Attention (MMA) layers with varying numbers of heads, which capture temporal dependencies with diverse levels of granularity. The accompanying illustration, as depicted in Figure 13, visually represents this architecture. A comparative analysis is conducted between MMA architecture and PMSA. As shown in Table 8, PMSA outperforms MMA in 26 out of 28 settings across a wide range of 7 datasets. It demonstrates that PMSA is more effective in capturing multi-scale dependency than MMA.

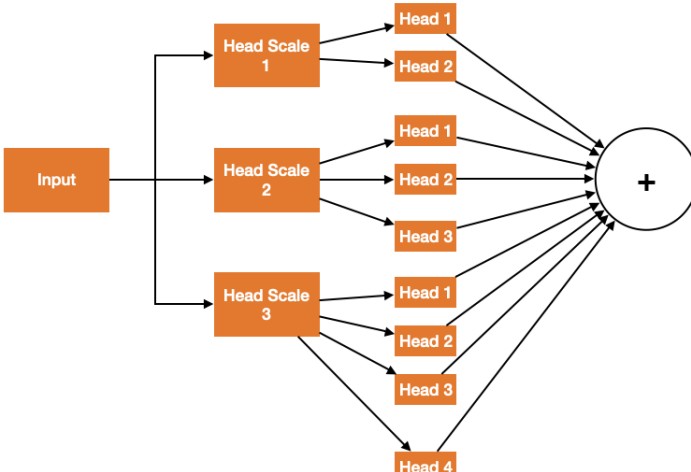

Figure 13: Illustration of multiscale with multiheads

Table 8: Empirical results of two multi-scale architectures:MMA and PMSA. Better results are in **bold**. The experiment uses the same setting as in Section A.1.4.

| Multi-scale architecture | | MMA | PMSA |
|---|---|---|---|
| Metrics | | MSE MAE | MSE MAE |
| ETTh1 | 96 | **0.367 0.394** | 0.373 0.394 |
| | 192 | **407 0.418** | 0.419 0.424 |
| | 336 | 0.426 0.434 | **0.420 0.428** |
| | 720 | 0.455 0.469 | **0.431 0.455** |
| ETTh2 | 96 | 0.276 0.337 | **0.276 0.337** |
| | 192 | 0.338 0.379 | **0.335 0.378** |
| | 336 | 0.330 0.385 | **0.326 0.384** |
| | 720 | 0.381 0.423 | **0.376 0.417** |
| ETTm1 | 96 | 0.289 0.341 | **0.286 0.338** |
| | 192 | 0.333 0.370 | **0.323 0.364** |
| | 336 | 0.366 0.393 | **0.358 0.385** |
| | 720 | 0.420 0.425 | **0.410 0.416** |
| ETTm2 | 96 | 0.173 0.281 | **0.162 0.249** |
| | 192 | 0.228 0.298 | **0.218 0.289** |
| | 336 | 0.289 0.339 | **0.275 0.326** |
| | 720 | 0.379 0.395 | **0.361 0.379** |
| Electricity | 96 | 0.130 0.224 | **0.128 0.221** |
| | 192 | 0.151 0.246 | **0.147 0.238** |
| | 336 | 0.166 0.261 | **0.164 0.256** |
| | 720 | 0.206 0.295 | **0.199 0.288** |
| Traffic | 96 | 0.383 0.262 | **0.371 0.255** |
| | 192 | 0.391 0.266 | **0.387 0.262** |
| | 336 | 0.411 0.274 | **0.400 0.268** |
| | 720 | 0.440 0.293 | **0.430 0.286** |
| Weather | 96 | 0.158 0.208 | **0.143 0.193** |
| | 192 | 0.199 0.245 | **0.187 0.236** |
| | 336 | 0.253 0.286 | **0.238 0.278** |
| | 720 | 0.320 0.335 | **0.314 0.333** |

