# OpenReview forum: "MGTST: Multi-scale and Cross-channel Gated Transformer for Multivariate long-term time-series forecasting"
_ICLR.cc/2024/Conference — Submitted to ICLR 2024_

### Official Review · Reviewer_oeni · 2023-10-29

**Soundness:** 2 fair
**Presentation:** 2 fair
**Contribution:** 2 fair
**Rating:** 3
**Confidence:** 3

**Summary:**

Based on the observation that existing transformer-based methods have issues, either overlooking the crucial interdependencies between channels or failing to capture various temporal patterns at different scales, this paper introduces a new method, MGTST (Multi-scale and cross-channel Gated Time-Series Transformer), which addresses these issues through the design of three modules:  Parallel Multi-Scale Architecture (PMSA), Temporal Embedding with Representation Tokens (TERT), and Cross-Channel Attention and Gated Mechanism (CCAGM). Additionally, the authors introduce the channel grouping strategy to alleviate channel interaction redundancy. Experiments are conducted on seven commonly used datasets to demonstrate the effectiveness of the proposed MGTST.

**Strengths:**

(1) The proposed method tackles issues of inadequate modeling of cross-channel dependencies and multi-scale temporal patterns in multivariate long-term time-series forecasting, which is technically sound.

(2) The authors conducted comparisons with several state-of-the-art baselines and ablation studies on different datasets to demonstrate the effectiveness of the proposed method.

(3) The paper has a clear structure and is generally easy to follow.

**Weaknesses:**

(1) The motivation of this paper is too general and lacks specific depth. The issues of mutual dependencies among channels and various temporal patterns have been long-standing. The authors fail to provide a deeper analysis and lack a detailed exposition of the specific problems the paper aims to address, and the reasons or new insights behind them.

(2) The claim that "One prominent limitation pertains to the inadequate modeling of cross-channel dependencies" is not fully convincing. The cross-dimension dependencies have been explored in (Zhang & Yan, 2022; Nie et al., 2022; Wu et al., 2022). It is not clear why or in which aspects the cross-dimension or cross-channel modeling of these methods is insufficient. The advantages of the proposed cross-dimension modeling module in this paper over these modeling methods are not clear. Similarly, for modeling multi-scale temporal patterns, the advantages of the corresponding module proposed in this paper over existing methods are also not clear.

(3) The method proposed in the paper seems like a simple stacking of modules. The authors could delve deeper into analyzing the relationship and interplay between the three proposed modules: PMSA, TERT, and CCAGM.

**Questions:**

(1) The motivation and some claims are not convincing enough. The advantages and relations among the proposed modules are not clear. Please see weaknesses for details.

(2) Hyperparameter settings. The paper conducted a hyperparameter analysis in the ablation study and provided additional hyperparameter configurations in the appendix. How do the authors determine the hyperparameters? For instance, the group size varies significantly across different datasets. Was it selected using a validation set?

(3) The main conceptual diagram in the article is too hastily drawn and occupies too much space.

---

> ### Author Response · Authors · 2023-11-15
> **Response to Reviewer oeni**
>
> Thanks for your questions.
>
> >**The motivation and some claims are not convincing enough. The advantages and relations among the proposed modules are not clear. Please see weaknesses for details.**
>
> The primary objective of our work is to develop a long-term time-series forecasting model that takes into account a broader range of information. The multi-scale architecture primarily focuses on incorporating multi-scale information, while the gated mechanism specifically addresses cross-channel information.
>
> The motivation for multi-scale architecture is that the optimal segment length varies with different forecasting lengths. We apply the Crossformer model to the ETTh1 dataset and present our findings below. For this analysis, we disable the Multi-Scale architecture to focus solely on the effect of segment length.
>
> **Forecasting length**&nbsp; &nbsp; **Segment length 6** &emsp; **Segment length 12** &emsp; **Segment length 24**
> ***
>
> 96 &emsp;&emsp;&emsp;&emsp;&emsp;&emsp;&emsp;&emsp;&emsp;&emsp;0.398 &emsp;&emsp;&emsp;&emsp;&emsp;&emsp;&emsp;0.397&emsp;&emsp;&emsp;&emsp;&emsp;&emsp;&emsp;&emsp;0.470
>
> 192&emsp;&emsp;&emsp;&emsp;&emsp;&emsp;&emsp;&emsp;&emsp;&ensp;0.649&emsp;&emsp;&emsp;&emsp;&emsp;&emsp;&emsp;&ensp;0.502&emsp;&emsp;&emsp;&emsp;&emsp;&emsp;&emsp;&emsp;0.423
>
> 336&emsp;&emsp;&emsp;&emsp;&emsp;&emsp;&emsp;&emsp;&emsp;&ensp;0.941&emsp;&emsp;&emsp;&emsp;&emsp;&emsp;&emsp;&ensp;0.598&emsp;&emsp;&emsp;&emsp;&emsp;&emsp;&emsp;&emsp;0.473
>
> 720&emsp;&emsp;&emsp;&emsp;&emsp;&emsp;&emsp;&emsp;&emsp;&ensp;0.787&emsp;&emsp;&emsp;&emsp;&emsp;&emsp;&emsp;&ensp;0.765&emsp;&emsp;&emsp;&emsp;&emsp;&emsp;&emsp;&emsp;0.509
>
> Based on the results, it is worthwhile to consider incorporating patches with different segment lengths into a unified model and subsequently generating the final prediction by evaluating the performance of each segment length.
>
> On the other hand, our further study posits that the gated mechanism is more robust with the penetration of noise regardless of forecasting length, however, other attention mechanisms such as Cross-channel attention and Cross-channel embedding adopted in Crossformer and Autoformer will suffer from noise. To commence our investigation, we initially train each model on the ETTh1 dataset. Subsequently, during the inference stage, we introduce Gaussian noise to six out of the seven channels and proceed to evaluate the forecasting accuracy of the final channels. The ensuing outcomes are presented as follows:
>
> **Forecasting Length**&nbsp; &nbsp;  **MGTST** **Crossformer** **Autoformer**
> ***
>
> 96 w/o noise &emsp; &emsp; &emsp; &ensp; 0.05215  0.06761 0.08748
>
> 96 w noise&emsp; &emsp; &emsp; &emsp; &emsp;0.05215 0.07246 0.10981
>
> 192 w/o noise &emsp; &emsp; &emsp; &nbsp; 0.06789 0.09117 0.13042
>
> 192 w noise&emsp; &emsp; &emsp; &emsp; &nbsp; 0.06789 0.10189 0.12685
>
> 336 w/o noise &emsp; &emsp; &emsp; &nbsp; 0.07802 0.18523 0.13585
>
> 336 w noise &emsp; &emsp; &emsp; &emsp; &nbsp;0.07801 0.10782 0.14064
>
> 720 w/o noise &emsp; &emsp; &emsp; &nbsp; 0.09770 0.53096 0.19737
>
> 720 w noise&emsp; &emsp; &emsp; &emsp; &nbsp; 0.09769 0.29152 0.11747
> ***
>
> The results show rapid performance degradation of Crossformer and Autoformer in long-term forecasting (336,720) with noise. On the other hand, the performance of the MGTST model exhibits only minimal variations across all forecasting horizons, indicating the robustness of the gated mechanism.
>
>
> >**Hyperparameter settings. The paper conducted a hyperparameter analysis in the ablation study and provided additional hyperparameter configurations in the appendix. How do the authors determine the hyperparameters? For instance, the group size varies significantly across different datasets. Was it selected using a validation set?**
>
>
> We determine the optimal hyperparameter configurations based on the model performance on the validation set. The determination of the group size was influenced by the limitations imposed by our experimental setup. Specifically, the group size needed to be a divisor of the total number of channels in the dataset in order to ensure compatibility. In the case of the Electricity dataset, the channel number could only be evenly divided by 107 and 3, resulting in the observed variation in the group size.
>
>
>
> >**The main conceptual diagram in the article is too hastily drawn and occupies too much space.**
>
> To optimize the utilization of space and enhance efficiency, we have resized the main conceptual diagram and integrated the gate mechanism diagram within it.

---

### Official Review · Reviewer_pwF5 · 2023-10-30

**Soundness:** 3 good
**Presentation:** 2 fair
**Contribution:** 2 fair
**Rating:** 5
**Confidence:** 3

**Summary:**

This paper introduces an interesting approach for long-term time series forecasting, harnessing the power of cross-channel attention and a multi-scale learning mechanism simultaneously. We enhance performance while reducing complexity by incorporating innovative temporal embedding mechanisms and a refined channel grouping strategy.

**Strengths:**

The paper is well-structured, offering a detailed description of each component's contribution.

The combined utilization of cross-channel attention and a multi-scale learning mechanism represents a fruitful approach in the provided scenarios.

Comprehensive comparison and ablation studies are provided, contrasting the proposed method with several existing approaches for long-term time series forecasting.

Overall, the motivation is clear.

**Weaknesses:**

The paper would benefit from providing more comprehensive details regarding the architecture and model settings of the transformer-based model.

To enhance the clarity of the figures, particularly Figures 2 and 3, consider adding explicit annotations and ensuring that the notations in the text align precisely with the corresponding elements in the figures.

Some areas that need attention include addressing missing notations (e.g., defining the meanings of "D").

In the case of Figure 2, it is crucial to clearly indicate which components represent the Temporal Embedding with Representation Tokens and which depict the Cross-Channel Attention and Gated Mechanism.

Further discussions on channel group sizes, scale numbers, and stride lengths would add depth and clarity to the paper.
Missing experiments on complex, multivariate, multimodal datasets, e.g., the air quality [1], interstate traffic [2]

**Questions:**

The use of a parallel architecture introduces substantial overhead when compared to the sequential approach. This overhead is expected to have a significant impact on prediction performance, particularly in the context of large-scale multimodal datasets. It's worth noting that the experimental section currently lacks the inclusion of these large-scale multimodal datasets for evaluation.

Considering the substantial computational overhead, the observed prediction improvement, such as the 1.5% reduction in Mean Squared Error (MSE), appears relatively minor.

The paper's evaluation is limited to simple time-series datasets, and it lacks assessment on more complex, multi-variate, and multi-modal datasets, e.g., the air quality [1], interstate traffic [2] (mentioned above in Weakness).

How does this approach compare with copula-based transformers [3] in the context of both long-term and short-term time-series prediction?

[1] Song Chen. Beijing Multi-Site Air-Quality Data. UCI Machine Learning Repository, 2019. DOI: https://doi.org/10.24432/C5RK5G.
[2] John Hogue. Metro Interstate Traffic Volume. UCI Machine Learning Repository, 2019. DOI: https://doi.org/10.24432/C5X60B.
[3] Alexandre Drouin, Etienne Marcotte, and Nicolas Chapados. TACTiS: Transformer-Attentional ´ Copulas for Time Series. In International Conference on Machine Learning, pp. 5447–5493. PMLR, 2022.

---

> ### Author Response · Authors · 2023-11-15
> **Response to Reviewer pwF5**
>
> Thanks for your questions.
>
> > **The use of a parallel architecture introduces substantial overhead when compared to the sequential approach. This overhead is expected to have a significant impact on prediction performance, particularly in the context of large-scale multimodal datasets. It's worth noting that the experimental section currently lacks the inclusion of these large-scale multimodal datasets for evaluation. Considering the substantial computational overhead, the observed prediction improvement, such as the 1.5% reduction in Mean Squared Error (MSE), appears relatively minor.**
>
> Despite introducing additional parallel architectures, MGTST could reduce the overall model size by reducing the size of the hidden layers in each parallel architecture while achieving better forecasting accuracy compared to traditional single-scale models. Herein, we illustrate this idea with a simple example by comparing the size of a single-scale model and a multi-scale model. Supposed that a single-scale model requires a hidden dimension size of 128 to achieve optimal forecasting results, however, a four-scale model can achieve the same or better level of accuracy with a hidden dimension size of only 24. Thus the overall model size of the four-scale model is still lower than the single scare model. A multiscale model such as the proposed MGTST can benefit from such parallel architecture to provide better prediction accuracy without introducing substantial overhead.
>
> To validate this conclusion, an experiment was conducted on the electricity dataset, comparing the forecasting results and total parameters between MGTST and PatchTST models. The experiment focused on a forecasting length of 96.
>
> **Parameter Size**&nbsp; &nbsp;  **PatchTST**&nbsp; &nbsp; **Parameter Size**&nbsp; &nbsp; **MGTST**
> ***
>
> 196530&emsp;&emsp;&emsp;&emsp;&emsp;0.137&emsp;&emsp;&emsp;&emsp;180630&emsp;&emsp;&emsp;&emsp;0.136
>
> 414322&emsp;&emsp;&emsp;&emsp;&emsp;0.135&emsp;&emsp;&emsp;&emsp;366465&emsp;&emsp;&emsp;&emsp;0.130
>
> 923634&emsp;&emsp;&emsp;&emsp;&emsp;0.132&emsp;&emsp;&emsp;&emsp;753301&emsp;&emsp;&emsp;&emsp;0.128
>
> The results confirm our conclusion that MGTST is both more efficient and more accurate.
>
>
> >**The paper's evaluation is limited to simple time-series datasets, and it lacks assessment on more complex, multi-variate, and multi-modal datasets, e.g., the air quality [1], interstate traffic [2] (mentioned above in Weakness)**
>
> In our research, we adhere to the standard dataset setting that has been used in previous related work, such as PatchTST [1] and Autoformer [2], for fair comparison. These datasets cover a wide range of fields, with different characteristics. varying size and channel size, which are sufficient to evaluate the performance of a forecasting method. While we recognize the importance of evaluating model performance on more datasets, we will leave this aspect for future research.
>
> >**How does this approach compare with copula-based transformers [3] in the context of both long-term and short-term time-series prediction?**
>
> After reviewing the copula-based transformer, we discovered that it is primarily designed for probabilistic forecasting tasks.  In contrast, MGTST and other baseline methods we considered in this work focus on addressing deterministic forecasting problems. As a result, direct comparisons between these two types of models may not be appropriate or meaningful due to the inherent differences in their forecasting approaches and objectives.

---

> > ### Comment · Reviewer_pwF5 · 2023-11-19
> > **Thank you**
> >
> > Thank you for your response. I have read your response and the other reviews and decided to keeo my score

---

### Official Review · Reviewer_PgoA · 2023-11-01

**Soundness:** 1 poor
**Presentation:** 2 fair
**Contribution:** 1 poor
**Rating:** 5
**Confidence:** 3

**Summary:**

**Summary:**

This paper introduces the MGTST (Multi-scale and cross-channel Gated Time-Series Transformer) model, aiming to address the limitations of current transformer-based models in multivariate long-term time-series forecasting. The authors propose several novel components, including Parallel Multi-Scale Architecture (PMSA), Temporal Embedding with Representation Tokens (TERT), Cross-Channel Attention and Gated Mechanism (CCAGM), and Channel Grouping (CG). The paper claims that MGTST outperforms existing models on several benchmark datasets.

**Strengths:**

A new MGTST model is proposed, which consists  PMSA, TERT, CCAGM, and CG modules.

**Weaknesses:**

*Insufficient Justification for Module Design:* The paper describes the procedures of the proposed modules but lacks a clear explanation of the underlying inductive biases and the rationale behind choosing these specific designs. For instance, the advantage of using gated attention over standard attention in time-series forecasting is not adequately discussed.

By the way, from my perspective, the parallel multi-scale structure and the CLS token seem not new in forecasting literature. For example, the multi-scale structure design in Crossformer starts from Unet. Unet contains the skip connection between the encoder and decoder at the same resolution which can be viewed as a parallel multi-scale design. Thus the Crossformer is not a simple sequential design as shown in Figure 3 but also contains a parallel design. The CARD (Xue et al., 2023) also considered the CLS token (e.g., Figure 1 in Xue et al., 2023). I have to admit that the exact design of the parallel multi-scale structure and CLS token are not the same as the counterparts in Crossformer and CARD but based on the current presentation I don't think they are significantly different.



*Numerical Comparison:* While the paper claims superior performance over baseline models, the results of PatchTST can be better according to the original paper. In PatchTST paper, results with two different input lengths, PatchTST/64 (512 input length) and PatchTST/42 (336 input length) are reported. In complexity settings (e.g., Electricity, Traffic, and Weather), PatchTST/64 with 512 input length has better performance than those with 336 input length. In particular, in the Traffic setting, PatchTST/64 obtains 0.360/0.379/0.392/0.432 MSE scores for 96/192/336/720 forecasting lengths, which are better than both MGTST-336 and MGTST-512. As authors consider two different input lengths, it would be better to also include experiments PatchTST/64.

*Dataloader Issue:* The sample codes seem to have the same issue in the test dataloader part as the original PatchTST paper, in which the last several test samples would be ignored. A similar observation has also been reported in TIDE paper. It would be better to fix the dataloader issues and update the crossposting results.



**Questions:**

1. Can authors elaborate more on why the gated mechanism is useful in the forecasting tasks?
2. Will the performance of the channel grouping module depend on the order of channels? After a random permutation on the order of channels, it seems that different groups will be obtained. Moreover, when setting group number 1, the settings of 5 datasets out of 7, how would channel grouping work, and how does it differ from standard attention?

**Minor Issues:**

1. In the paragraph before section 3.1 and section 3.2, $\frac{L\_{input} - L\_{patch}}{S_{stride}}$ -->  $\lfloor\frac{L\_{input} - L\_{patch}}{S_{stride}}\rfloor$

2. Discussion for Crossformer in Section A.1.3. The Crossformer and PatchTST are concurrent work. The statement that "*...adds a cross-attention layer on the top of PatchTST*" seems improper from my perspective.


At this stage, the paper's contributions do not appear substantial enough for acceptance at a top-tier machine learning conference like ICLR. The novelty and effectiveness of the proposed components need to be more convincingly demonstrated. However, I am open to reconsidering my decision if the authors can address these concerns in their rebuttal.




**Reference**

TIDE: Das, Abhimanyu, Weihao Kong, Andrew Leach, Rajat Sen, and Rose Yu. "Long-term Forecasting with TiDE: Time-series Dense Encoder

**Strengths:**

Please refer to the Strengths section in Summary.

**Weaknesses:**

Please refer to the Weaknesses section in Summary.

**Questions:**

Please refer to the Questions section in Summary.

---

> ### Author Response · Authors · 2023-11-15
> **Response to Reviewer PgoA**
>
> Thanks for your questions.
>
> > **Can authors elaborate more on why the gated mechanism is useful in the forecasting tasks?**
>
> As we demonstrate in the paper that gated can improve the overall performance. Besides, our further study posits that the gated mechanism is more robust with the penetration of noise regardless of forecasting length, however, other attention mechanisms such as Cross-channel attention and Cross-channel embedding adopted in Crossformer and Autoformer will suffer from noise. To commence our investigation, we initially train each model on the ETTh1 dataset. Subsequently, during the inference stage, we introduce Gaussian noise to six out of the seven channels and proceed to evaluate the forecasting accuracy of the final channels. The ensuing outcomes are presented as follows:
>
> **Forecasting Length**&nbsp; &nbsp;  **MGTST** **Crossformer** **Autoformer**
> ***
>
> 96 w/o noise &emsp; &emsp; &emsp; &ensp; 0.05215  0.06761 0.08748
>
> 96 w noise&emsp; &emsp; &emsp; &emsp; &emsp;0.05215 0.07246 0.10981
>
> 192 w/o noise &emsp; &emsp; &emsp; &nbsp; 0.06789 0.09117 0.13042
>
> 192 w noise&emsp; &emsp; &emsp; &emsp; &nbsp; 0.06789 0.10189 0.12685
>
> 336 w/o noise &emsp; &emsp; &emsp; &nbsp; 0.07802 0.18523 0.13585
>
> 336 w noise &emsp; &emsp; &emsp; &emsp; &nbsp;0.07801 0.10782 0.14064
>
> 720 w/o noise &emsp; &emsp; &emsp; &nbsp; 0.09770 0.53096 0.19737
>
> 720 w noise&emsp; &emsp; &emsp; &emsp; &nbsp; 0.09769 0.29152 0.11747
> ***
> The results show rapid performance degradation of Crossformer and Autoformer in long-term forecasting (336,720) with noise. On the other hand, the performance of the MGTST model exhibits only minimal variations across all forecasting horizons, indicating the robustness of the gated mechanism.
>
> > **Will the performance of the channel grouping module depend on the order of channels? After a random permutation on the order of channels, it seems that different groups will be obtained. Moreover, when setting group number 1, the settings of 5 datasets out of 7, how would channel grouping work, and how does it differ from standard attention?**
>
> MGTST selectively applies channel grouping solely to datasets that possess a substantial number of channels. Despite its effectiveness, this approach does not yield significant performance alterations. To illustrate this, we present the results of MGTST with varying grouping sizes in the Electricity dataset with forecasting length of 720:
>
> **Grouping size**&nbsp; &nbsp;  **MGTST**
> ***
> 107&emsp;&emsp;&emsp;&emsp;&emsp;&emsp;0.200
>
> 3&emsp;&emsp;&emsp;&emsp;&emsp;&emsp;&emsp;0.202
>
> 1&emsp;&emsp;&emsp;&emsp;&emsp;&emsp;&emsp;0.204
> ***
>
> for datasets such as ETTh1, which contain only seven channels, additional grouping of channels is unnecessary. In such cases, the standard attention mechanism is employed over representation tokens to achieve optimal performance
>
> > **Rationale behind parallel multi-scale architecture?**
>
> We observed that the optimal segment length varies with different forecasting lengths. We apply the Crossformer model to the ETTh1 dataset and present our findings below. For this analysis, we disable the Multi-Scale architecture to focus solely on the effect of segment length.
>
> **Forecasting length**&nbsp; &nbsp; **Segment length 6** &emsp; **Segment length 12** &emsp; **Segment length 24**
> ***
>
> 96 &emsp;&emsp;&emsp;&emsp;&emsp;&emsp;&emsp;&emsp;&emsp;&emsp;0.398 &emsp;&emsp;&emsp;&emsp;&emsp;&emsp;&emsp;0.397&emsp;&emsp;&emsp;&emsp;&emsp;&emsp;&emsp;&emsp;0.470
>
> 192&emsp;&emsp;&emsp;&emsp;&emsp;&emsp;&emsp;&emsp;&emsp;&ensp;0.649&emsp;&emsp;&emsp;&emsp;&emsp;&emsp;&emsp;&ensp;0.502&emsp;&emsp;&emsp;&emsp;&emsp;&emsp;&emsp;&emsp;0.423
>
> 336&emsp;&emsp;&emsp;&emsp;&emsp;&emsp;&emsp;&emsp;&emsp;&ensp;0.941&emsp;&emsp;&emsp;&emsp;&emsp;&emsp;&emsp;&ensp;0.598&emsp;&emsp;&emsp;&emsp;&emsp;&emsp;&emsp;&emsp;0.473
>
> 720&emsp;&emsp;&emsp;&emsp;&emsp;&emsp;&emsp;&emsp;&emsp;&ensp;0.787&emsp;&emsp;&emsp;&emsp;&emsp;&emsp;&emsp;&ensp;0.765&emsp;&emsp;&emsp;&emsp;&emsp;&emsp;&emsp;&emsp;0.509
>
> Based on the results, it is worthwhile to consider incorporating patches with different segment lengths into a unified model and subsequently generating the final prediction by evaluating the performance of each segment length.
>
> > **Difference between CLS token of MGTST and CARD (Xue et al., 2023)?**
>
> The CLS token in CARD is used for static variables, which are manually generated and not initialized. Its purpose is to enrich the input information, similar to the function of Attributes and Dynamic Covariates in TIDE (Das et al., 2023). On the other hand, the CLS token in MGTST is initialized and updated through self-attention to represent the entire channel. This aligns more closely with the original definition of CLS token in Vision transformer.  To be best of our knowledge, this is the first time that CLS tokens are introduced in multivariate time-series forecasting tasks for channel interaction modelling.

---

### Meta-Review · Area_Chair_mJBT · 2023-12-07

**Metareview:**

The reviewers raised multiple concerns, including the novelty representation of the cross-channel idea, baselines and experimental setting.  After the rebuttal, some concerns have not been resolved yet, which suggests a reject.

**Justification For Why Not Higher Score:**

After the rebuttal, some concerns have not been resolved yet, which suggests a reject.

**Justification For Why Not Lower Score:**

n/a

---

### Decision · Program_Chairs · 2024-01-16

Reject